computational chemistry/physical chemistry

marine compounds, cancer, virtual screening, molecular dynamics, replica exchange molecular dynamics, free energy

**Author for correspondence:**
Son Tung Ngo
e-mail: ngosontung@tdtu.edu.vn

This article has been edited by the Royal Society of Chemistry, including the commissioning, peer review process and editorial aspects up to the point of acceptance.

# Marine derivatives prevent *w*MUS81 *in silico* studies

Son Tung Ngo[1,2], Khanh B. Vu[3,4], Minh Quan Pham[5,6],
Nguyen Minh Tam[2] and Phuong-Thao Tran[7]

[1]Laboratory of Theoretical and Computational Biophysics, Ton Duc Thang University, Ho Chi Minh City, Vietnam
[2]Faculty of Applied Sciences, Ton Duc Thang University, Ho Chi Minh City, Vietnam
[3]Department of Chemical Engineering, International University, Ho Chi Minh City, Vietnam
[4]Vietnam National University, Ho Chi Minh City, Vietnam
[5]Institute of Natural Products Chemistry, Vietnam Academy of Science and Technology, Hanoi, Vietnam
[6]Graduate University of Science and Technology, Vietnam Academy of Science and Technology, Hanoi, Vietnam
[7]Department of Pharmaceutical Chemistry, Hanoi University of Pharmacy, Hanoi, Vietnam

STN, 0000-0003-1034-1768; KBV, 0000-0001-5875-2481;
MQP, 0000-0001-6922-1627; NMT, 0000-0003-3153-4606;
P-TT, 0000-0003-4855-2544

The winged-helix domain of the methyl methanesulfonate and ultraviolet-sensitive 81 (*w*MUS81) is a potential cancer drug target. In this context, marine fungi compounds were indicated to be able to prevent *w*MUS81 structure via atomistic simulations. Eight compounds such as **D197** (*Tryptoquivaline U*), **D220** (*Epiremisporine B*), **D67** (*Aspergiolide A*), **D153** (*Preussomerin G*), **D547** (*12,13-dihydroxyfumitremorgin C*), **D152** (*Preussomerin K*), **D20** (*Marinopyrrole B*) and **D559** (*Fumuquinazoline K*) were indicated that they are able to prevent the conformation of *w*MUS81 via forming a strong binding affinity to the enzyme via perturbation approach. The electrostatic interaction is the dominant factor in the binding process of ligands to *w*MUS81. The residues Trp55, Arg59, Leu62, His63 and Arg69 were found to frequently form non-bonded contacts and hydrogen bonds to inhibitors. Moreover, the influence of the ligand **D197**, which formed the lowest binding free energy to *w*MUS81, on the structural change of enzyme was investigated using replica exchange molecular dynamics simulations. The obtained results indicated that **D197**, which forms a strong binding affinity, can modify the structure of *w*MUS81. Overall, the marine compounds probably inhibit *w*MUS81 due to forming a strong binding affinity to the enzyme as well as altering the enzymic conformation.

## 1. Introduction

Methyl methanesulfonate ultraviolet-sensitive gene clone 1 (MUS81) is a member of the Xpf family of structure-specific DNA endonucleases, which play overlapping and essential roles in the completion of

homologous recombination [1]. Human MUS81 (hMUS81) nicked Holliday junctions and D-loops, replication forks and 3′ flap substrates *in vitro*, suggesting a number of possible *in vivo* functions [2]. The winged-helix domain at the N-terminus of MUS81 (*w*MUS81) mutations was indicated to reduce the binding of the DNA substrates and modulate the endonuclease activity of the MUS81 complex [3] leading to a crucial role of MUS81 in DNA replication, repair and transcription [4]. Inhibitions of MUS81 improve the chemical sensitivity of various anti-cancer drugs, such as 5-fluorouracil, camptothecin, olaparib and cisplatin by different mechanisms in diverse cancer cells. The downregulation of MUS81 enhanced the sensitivity to camptothecin and olaparib in serous ovarian cancer cell lines and xenograft model [5,6], and also played a role as a potential marker for the malignancy of gastric cancer [7]. The inhibition of MUS81 by siRNA reinforced the sensitiveness to 5-fluorouracil in breast carcinoma cell lines [8]. MUS81 knockdown increased the chemosensitivity of hepatocellular carcinoma cells and colon cancer cells by activating S-phase arrest and elevating apoptosis [9,10]. The lack of MUS81 could improve the sensitivity of cisplatin as well as terminate the enlargement and intension of serous ovarian cancer [11]. Moreover, MUS81 inhibition enhanced the sensitivity to anti-tumour in epithelial ovarian cancer via regulating CyclinB pathway as well [4]. Thus, MUS81 could be presented as a novel molecular target for future anti-cancer treatment. Accordingly, MUS81 inhibitors that specifically bind to MUS81 might enhance the chemosensitivity in the cancer cells leading to improving the efficiency of therapeutic treatment.

Currently, designing novel therapy using natural compounds, which are extracted from herbs and foods, is interestingly executed since these compounds normally adopt fewer side effects and toxics than synthetic ones [12–15]. The natural compounds are also highly available for large production. Several compounds with high antioxidant and anti-inflammatory properties were found to be able to remodel or inhibit the biological activities of a protein via various pathways [16–18]. The inhibition probably activates through forming hydrogen bond (HB), adopting hydrophobic contacts and interfering with π–π stacking interaction with the active site residues of the enzyme [19,20]. Moreover, marine compounds, which are extracted from marine organisms, provide widely biological activities and high availability that they emerge as highly candidates for drug screening [21–26].

Computer-aided drug design (CADD) is regularly employed to rapidly discover potent compounds, which are able to inhibit the biological and chemical functions of an enzyme [27–31]. Therapeutic development is thus accelerated. Particularly, the ligand-binding free energy, $\Delta G$, is regularly calculated since characterizing to the experimental inhibition constant, $K_i$. In computation, the $\Delta G$ can be characterized via molecular docking and/or molecular dynamics (MD) and/or quantum chemical (QM) simulations. Accurately estimating the value is a very essential problem to characterize which ligand can bind to a receptor or not [32–34]. Therefore, in this context, AutoDock Vina (Vina) [35,36] was carried out to preliminarily predict the *w*MUS81-ligands binding poses and affinities since it is known as one of the most powerful docking techniques [37–39]. The ligand-binding poses to *w*MUS81 were then concentrated using MD simulations. Moreover, the ligand-binding affinities were yielded to refine via the double-annihilation binding free energy (FEP) approach [33,40]. A shortlist of marine compounds, which can bind well to *w*MUS81, was thus obtained. Furthermore, the physical insights into the *w*MUS81-inhibitor binding were also clarified. In addition, the stronger binder normally adopts a strong effect on the protein, resulting in the structural change of protein was recorded [41]. Therefore, the influence of the ligand, which formed the strongest binding affinity to the *w*MUS81, on the structure of *w*MUS81 was investigated using temperature replica exchange molecular dynamics (REMD) simulations [42]. The obtained outcomes possibility yield to enhance cancer therapy and guide for further computational studies.

# 2. Material and methods

## 2.1. Structure of *w*MUS81 and marine derivatives

The atomistic details of *w*MUS81 were taken from the Protein Data Bank with the identity of 2MC3 [3]. The three-dimensional structure of 502 ligands were labelled from **D1** to **D502**, which were downloaded from the PubChem database [43] according to previous works [44–46]. The other ligands, which were labelled from **D503** to **D563**, were obtained via experimental investigations on marine fungi samples, and involve *Aspergillus* sp. and *Penicillium* sp., referring to the recent work [26].

## 2.2. Molecular docking simulations

The preliminary estimations of the ligand-binding poses and affinities were carried out by using AutoDock Vina (Vina) [47], which is known as one of the most common docking approaches [37].

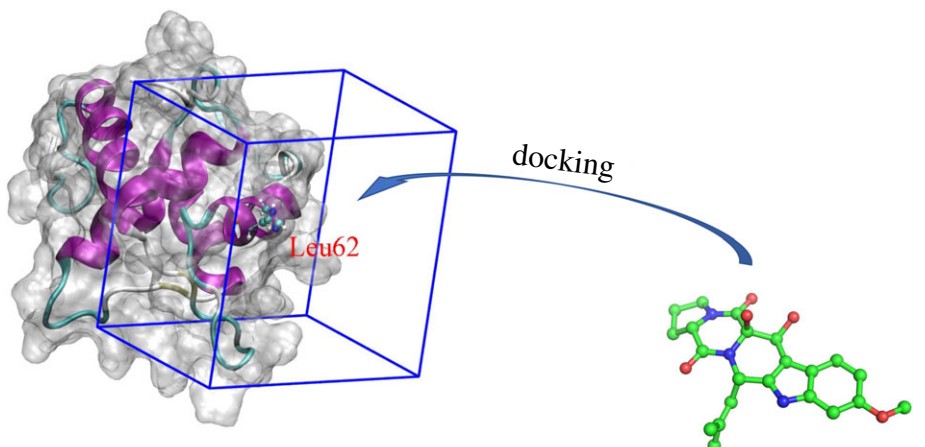

**Figure 1.** Docking modelling simulations. Ligands were docked to the binding region, which was embraced by rectangular box with a volume of 13.52 nm$^3$.

The AutoDockTools 1.5.6 was employed to topologize for both receptor and ligands [48]. The grid centre was selected as the residue Leu62 sidechain centre of mass (cf. figure 1) referring to the recent work [49]. The grid size was $2.6 \times 2.6 \times 2.0$ nm (figure 1). The docking parameter, exhaustiveness, was chosen as 8 referred to the previous work [37]. During the docking simulations, the ligand was flexible but the receptor was fully rigid.

## 2.3. Molecular dynamics simulations

GROMACS 2019.4 using graphics processing unit (GPU) computing was employed to simulate the dynamics of the *w*MUS81-inhibitor system in solution [50]. The complex was topologized according to the recent studies [49,51]. In particular, the *w*MUS81/ions and water molecules were parametrized via the Amber99SB-ILDN force field [52] and the TIP3P water model [53], respectively. Besides that, the inhibitor was parametrized using the general Amber force field (GAFF) [54] using a combination of AmberTools18 [55] and ACPYPE [56] packages. Ligand atomic charges were computed via the restrained electrostatic potential (RESP) method [57] using quantum mechanical calculations with the level of theory at B3LYP/6-31G(d,p). The implementation of the force fields is a suitable selection for free energy determination [58,59]. The *w*MUS81 complex was placed in a dodecahedron box with a size of $7.0 \times 7.0 \times 7.0$ nm (cf. figure 2), being measured with a periodic boundary condition [60]. Four Na$^+$ ions were approximately used to neutralize the system. Overall, the *w*MUS81 system consists of *ca* 25 000 atoms. Moreover, the system comprising the ligand in solution was also prepared and simulated. The size of the ligand in solution was *ca* 74 nm$^3$ comprising *ca* 7200 atoms in total.

The MD parameters were referred to the prior works [41,61]. In particular, the non-bonded pair between surrounding atoms is calculated within a radius of 9 Å with the pair list is reorganized every 5 ps during the MD simulations. The MD integral was performed with 2 fs timestep. The van der Waals (vdW) interaction is affected within a cut-off of 9 Å. The coulombic interaction is implemented via the particle mesh Ewald method [62] within a non-bonded pair radius. The MD temperature is 310 K. The complex was equilibrated via three steps as followed as the energic minimization, canonical (NVT) simulations and isobaric–isothermal (NPT) simulations. The solvated complex and ligand systems were then mimicked over 20 and 5 ns of MD simulations, respectively. The simulations were repeated two times.

## 2.4. Replica exchange molecular dynamics simulations

The REMD simulations were manipulated to investigate the structural change of the *w*MUS81 entire phase space when the inhibitor is present and absent. During the simulations, 32 replicas were simultaneously mimicked in various temperatures ranging from 310.00 to 414.26 K, which the list was suggested by a webserver [63] as well as 310.00, 313.02, 316.06, 319.13, 322.21, 325.32, 328.44, 331.60, 334.77, 337.97, 341.18, 344.43, 347.69, 350.97, 354.28, 357.62, 360.97, 364.35, 367.76, 371.19, 374.63, 378.11, 381.61, 385.14, 388.69, 392.26, 395.86, 399.49, 403.15, 406.83, 410.54 and 414.26 K. The REMD

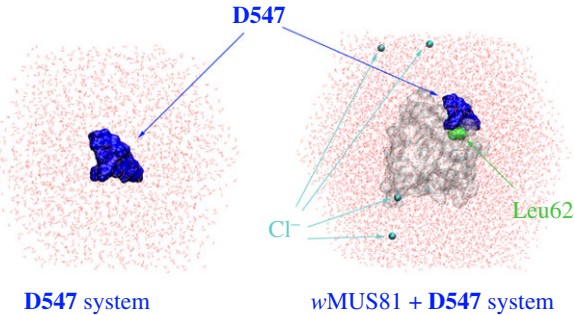

**D547**

Cl⁻

Leu62

**D547** system *w*MUS81 + **D547** system

**Figure 2.** The MD conformations of the ligand **D547** and *w*MUS81+**D547** systems in solution.

simulations were carried out with a length of 100 ns each replica (6400 ns for two systems in total). The acceptance ratio was 15% approximately. Therefore, the *w*MUS81 system easily escapes local minima due to exchanging shapes with higher temperature copies. The exchange was attempted every 1 ps if the exchange probability between adjacent replicas satisfies the Metropolis criterion [33]. The inputs of REMD simulations are the equilibrated conformations of isolated *w*MUS81 or *w*MUS81+inhibitor in solution, which were the final structures of NPT simulations.

## 2.5. Analysis tools

The protonation states of inhibitors were predicted using a tool of the ChemAxon protocol as well as chemicalize (chemicalize was used for the prediction of chemical properties; see https://chemicalize.com/, developed by ChemAxon). The non-bonded contact between ligands and residual elements of *w*MUS81 was totalized when the pair between non-hydrogen atoms is smaller than 4.5 Å. Two-dimensional interaction diagrams were prepared via the free Maestro [64]. The binding free energy between inhibitors and *w*MUS81 was determined via the double-annihilation binding free method [33,40,65]. The details of FEP approach were reported in the previous work [33]. In particular, the coupling parameter $\lambda$ was used to modify the Hamiltonian of the system from *bound* state to *unbound* state. The free energy change of the process was summed using the Bennet acceptance ratio method [66]. The $\lambda$-modification simulations were thus performed to calculate the free energy change of ligand-annihilation from the solvated complex and ligand systems. The difference of free energy change of the two processes is the binding free energy of ligands to receptors. Eight values of $\lambda$, used to change electrostatic interaction, are 0.00, 0.10, 0.20, 0.35, 0.50, 0.65, 0.80 and 1.00. Nine values of $\lambda$, used to change vdW interaction, are 0.00, 0.10, 0.25, 0.35, 0.50, 0.65, 0.75, 0.90 and 1.00. The free energy landscape (FEL) was built via the GROMACS tool [50] with two reaction coordinates being the first and the second eigenvectors estimated by the principal component analysis (PCA) method [67].

# 3. Results and discussion

## 3.1. Molecular docking simulation

As mentioned above, molecular docking simulations were performed to preliminarily evaluate the ligand-binding poses and affinities to *w*MUS81. The outcomes are shown in table 1 and electronic supplementary material, table S1. Here, Vina, a free and popular docking package with a successful-docking rate of up to 81% [35–37], was executed to find binding poses and affinities of 563 marine compounds to receptor referring to the recent work [49]. The docking energies range from −2.8 to −7.4 kcal mol⁻¹ with a median of −5.17 ± 0.63 kcal mol⁻¹, the calculated error is the standard deviation. The distribution of docking energies is thus presented in figure 3. According to the docking results, 3% top-lead compounds, having $\Delta G_{\text{Vina}} < -6.0$ kcal mol⁻¹, possibly bind to *w*MUS81 with a large amount of free energy difference of binding. Although, as mentioned above, Vina approach regularly adopts appropriate results, the affinity of 16 compounds to *w*MUS71 would be further evaluated using MD/REMD simulations. The strategy is normally applied since the docking simulations implement many approximations to enhance the calculation performance such as a rigid receptor, a limited number of trial ligand positions, employed united-atoms force field, using implicit solvent model, etc.

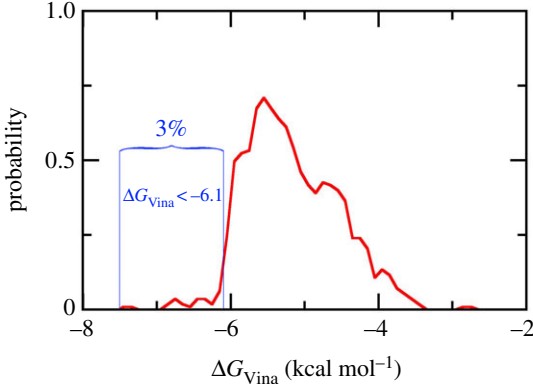

**Figure 3.** The distribution of docking energies between 563 marine compounds and $w$MUS81, which were obtained via Vina protocol.

**Table 1.** The binding free energy obtained from the docking and perturbation simulations.[a]

| no. | inhibitors | $\Delta G_{Docking}$ | $\Delta G_{cou}$ | $\Delta G_{vdW}$ | $\Delta G_{FEP}$ |
|---|---|---|---|---|---|
| 1 | **D197** | −7.4 | −12.23 | −6.60 | −18.83 ± 0.91 |
| 2 | **D220** | −6.8 | −10.30 | −7.38 | −17.69 ± 4.20 |
| 3 | **D67** | −6.4 | −10.37 | −5.33 | −15.70 ± 0.47 |
| 4 | **D153** | −6.7 | −11.18 | −2.51 | −13.70 ± 0.12 |
| 5 | **D547** | −6.1 | −1.28 | −11.35 | −12.63 ± 0.47 |
| 6 | **D152** | −6.7 | −5.22 | −5.64 | −10.86 ± 1.98 |
| 7 | **D20** | −6.8 | −4.68 | −4.90 | −9.57 ± 3.78 |
| 8 | **D559** | −6.1 | −2.17 | −5.97 | −8.14 ± 0.43 |
| 9 | **D87** | −6.1 | −0.26 | −3.40 | −3.65 ± 0.12 |
| 10 | **D548** | −6.3 | 4.14 | −7.29 | −3.15 ± 0.13 |
| 11 | **D542** | −6.2 | 2.81 | −5.96 | −3.14 ± 0.39 |
| 12 | **D545** | −6.5 | 3.61 | −6.05 | −2.45 ± 0.57 |
| 13 | **D561** | −6.1 | 3.02 | −4.76 | −1.74 ± 0.55 |
| 14 | **D554** | −6.1 | 1.16 | −2.43 | −1.27 ± 0.22 |
| 15 | **D560** | −6.4 | 8.83 | −9.70 | −0.87 ± 0.40 |
| 16 | **D550** | −6.1 | 2.21 | −2.87 | −0.66 ± 0.03 |

[a]The unit is kcal mol$^{-1}$.

As mentioned above, the ligand-binding affinity of the 16 top-lead compounds to $w$MUS81 would be re-evaluated using MD simulations (table 1) due to some limitations of the docking approach as mentioned above. The outcome of Vina is an appropriate result since forming a good correlation with the FEP calculations (cf. figure 4), which approach is known as the most accurate technique [28]. The obtained Pearson correlation is $R_{FEP}^{Vina} = 0.68 \pm 0.17$, in which the computed error was estimated via the bootstrapping method with 1000 rounds of calculations [68]. It may argue that Vina is a suitable solution for explorative determining ligand-binding affinity to $w$MUS81 quickly.

Two-dimensional interaction diagrams between inhibitors and $w$MUS81 were prepared using the free Maestro package [64] and the results are shown in figure 5 and electronic supplementary material, table S3. Eight compounds shown in figure 5 are potential inhibitors for $w$MUS81 via MD-refined simulations (see below). Over these diagrams, it may argue that the residues Arg59, Ser60, Leu62, His63, Asn65, Val67, Leu68 and Arg69 play a critical role in the binding process of ligands to $w$MUS81. Especially, the residues Arg59, His63 and Arg69 not only form HB contacts with the inhibitors but also adopt π-cation/ π–π contacts to the ligands.

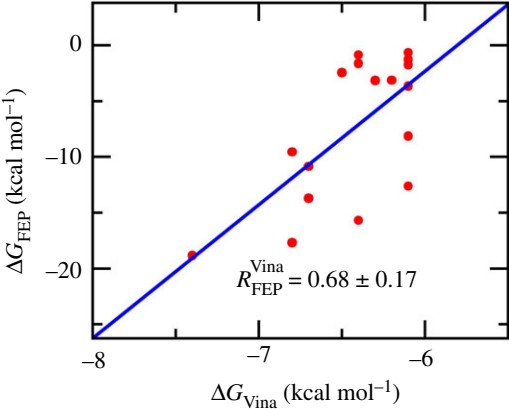

**Figure 4.** Association of molecular docking and FEP calculations. The error was computed via 1000 rounds of bootstrapping method.

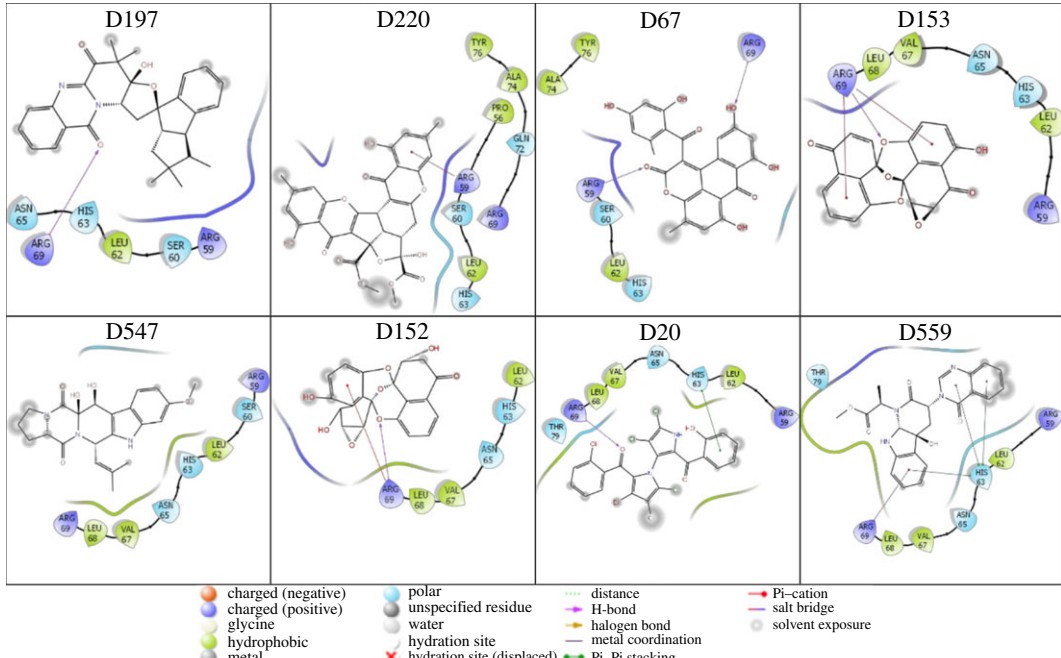

**Figure 5.** Two-dimensional interaction diagram between inhibitors and *w*MUS81. The diagram was provided from the analysis of the docking pose between ligands and *w*MUS81.

## 3.2. Molecular dynamics-refined simulations

As mentioned above, the molecular docking simulations were performed using many constraints; the MD simulations were thus manipulated to improve the obtained outcomes [30,69]. The binding poses of ligands to *w*MUS81 were used as initial conformations for mimicking complex by GROMACS 2019.4 [50]. The ligand-binding free energy was then assessed via perturbation simulations [33]. In this context, the MD simulations were carried out with a length of 20 and 5 ns for complexed *w*MUS81-inhibitor and isolated inhibitor in solution, correspondingly. Particularly, almost systems gained the stabilized conformations after half of the MD trajectory (cf. electronic supplementary material, table S2).

The intermolecular non-bonded contacts and HBs between receptor residues and inhibitors were probed in order to predict the binding mechanism of ligands to *w*MUS81. The investigations were carried out over the equilibrium structures of the solvated complexes. The obtained results are described in electronic supplementary material, figure S1, which presents 77 residues forming non-bonded contacts with inhibitors over 5% of the considered structures (34 000 snapshots in total). Among them, 44 residues adopt HBs to inhibitors. The obtained results suggested that inhibitors probably moved the entire surface of *w*MUS81 during MD simulations, confirming that the simulations were unbiased to any

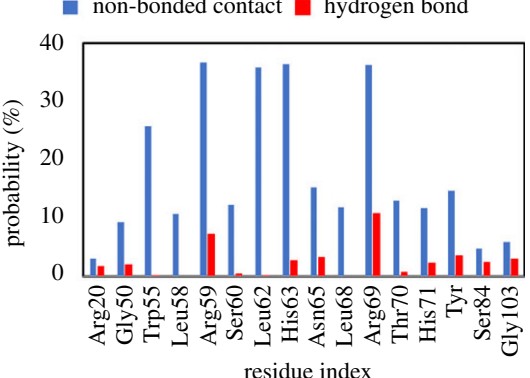

**Figure 6.** Pivotal residues of *w*MUS81 adopting intermolecular non-bonded contacts and HBs to inhibitors. The outcomes were obtained over equilibrium structures via MD simulations of all complexes.

specific conformation. In order to characterize which are the most pivotal residues, we have counted residues that formed more than 10% non-bonded contacts or 2% HBs to inhibitors. Sixteen residues, which are shown in figure 6, were satisfied with the criteria. We may conclude that the residues Trp55, Arg59, Leu62, His63 and Arg69 are important residues controlling the ligand-binding free energy of ligands to *w*MUS81 since these residues form at least 26% non-bonded contacts to trial ligands. Moreover, among five residues, both residues Arg59 and Arg69 dominated over the other residues since forming both non-bonded contacts and HBs to inhibitors (cf. figure 6). Furthermore, probable mutations at these residues could significantly alter the ligand-binding affinity to *w*MUS81.

## 3.3. Perturbation simulations

In the recent report [49], the FEP calculations fruitfully evaluated the ligand-binding affinity of ligands to *w*MUS81 as well as the highest accurate binding free energy approaches [27,28]. Therefore, in this context, we used FEP simulations to refine the binding free energy of inhibitors to *w*MUS81. In particular, the equilibrated shapes of complexes and ligands in solution at 20 and 5 ns were used as initial structures of $\lambda$-alteration simulations, respectively. The obtained results are reported in table 1, in which the $\Delta G_{\text{FEP}}$ falls in the range from −0.66 to −18.83 kcal mol$^{-1}$ with a median of *ca* −7.39 kcal mol$^{-1}$. Eight compounds were indicated by FEP to form binding affinity larger than −8.0 kcal mol$^{-1}$, which would respect to an inhibition constant $K_i$ in the micromolar range or smaller (table 1). Among these compounds, five inhibitors adopted a large binding affinity to *w*MUS81 with the respective $K_i$ in the subnanomolar range. Therefore, we may argue that five compounds named **D197**, **D220**, **D67**, **D153** and **D547** are highly potent inhibitors preventing the conformation of *w*MUS81. Besides, three compounds, **D152**, **D20** and **D559**, are also good candidate ones. Moreover, on average over all compounds, the $\Delta G_{\text{cou}}$ and $\Delta G_{\text{vdW}}$ values form a median of *ca* −1.58 and −5.82 kcal mol$^{-1}$, correspondingly. However, when considering over eight top-lead compounds, the $\Delta G_{\text{cou}}$ and $\Delta G_{\text{vdW}}$ values form a median of *ca* −7.18 and −6.21 kcal mol$^{-1}$, respectively. The electrostatic free energy of binding dominates over the vdW one in controlling the binding process of ligands to *w*MUS81. It is well consistent with the observation in docking analysis (figure 5).

## 3.4. Replica exchange molecular dynamics simulations

It should be noted that the stronger ligand-binding to protein means better inhibition [28]. Moreover, the stronger binder normally adopts a strong effect on the protein, resulting in the structural change of protein being recorded [41]. Therefore, in this context, the influence of the ligand **D197** on the structure of *w*MUS81 was investigated using REMD simulations since the ligand formed the largest binding affinity to *w*MUS81 via FEP calculations. In particular, the structural change of *w*MUS81 over the REMD simulations [42] were performed with a length of 100 ns each. The conformational ensemble of *w*MUS81 in the presence of the ligand **D197** and absence of the ligand was probed. The first 40 ns of REMD simulations was removed from any analysis to evade the initial bias. Figure 7 describes the FELs and representative structures of *w*MUS81 over two systems. The FEL was produced by two principal components, which were obtained from the PCA analysis. Absolutely, the ligand significantly influences

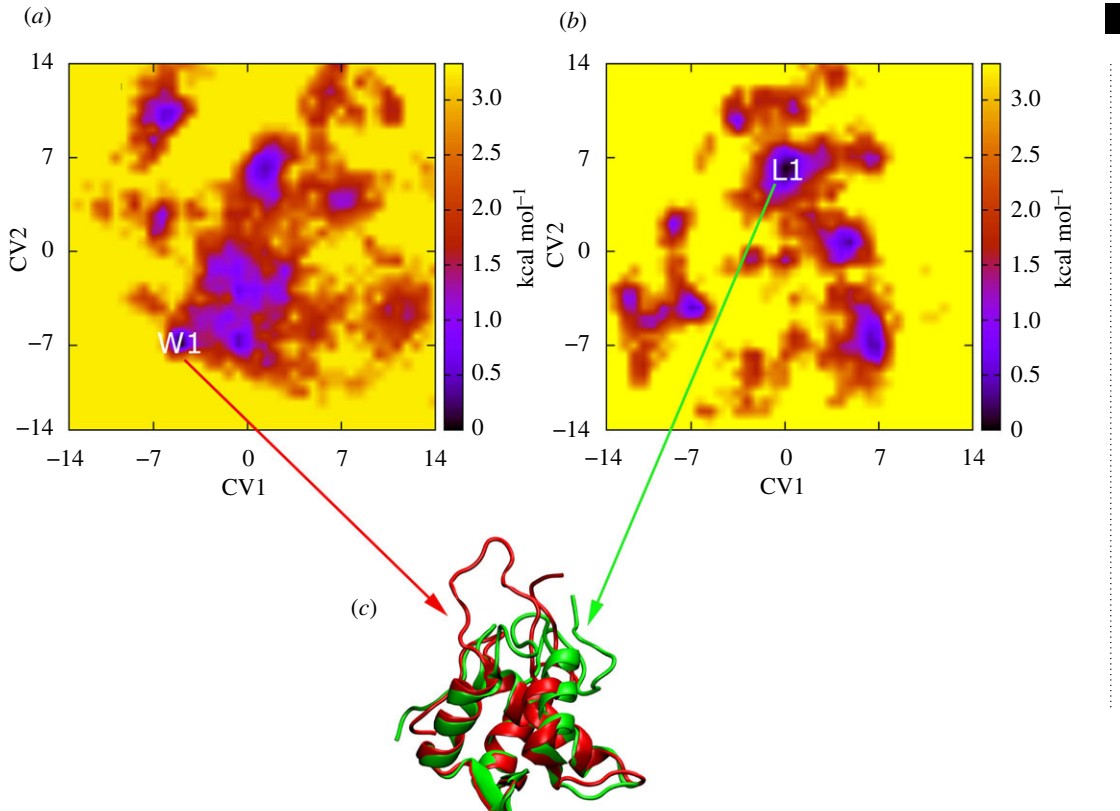

**Figure 7.** The FEL of *w*MUS81 (*a*) and *w*MUS81+**D197** (*b*) constructed by the two principal components attained from the PCA analysis. The representative structure W1 of *w*MUS81 without inhibitor was shown via red colour in (*c*), in which the *w*MUS81 adopted α-structure at sequences 10–23, 33–39, 54–64 and 80–91; β-sheet at sequences 31–32, 67–70 and 75–78. The representative structure L1 of *w*MUS81 in the present of D197 was described using green colour in (*c*), in which, *w*MUS81 formed α-structure at sequences 10–23, 33–43, 50–64 and 80–87; β-sheet at sequences 31–32, 67–70 and 75–78.

the FEL of *w*MUS81 and also alters the optimized structures of the enzyme. Although the β-sheet regions of *w*MUS81 were unchanged, the α-structure domains were rigidly impacted (figure 7). In particular, without the presence of the ligand, *w*MUS81 forms α-structure at sequences 10–23, 33–39, 54–64 and 80–91, while the corresponding sequences of *w*MUS81+**D197** are 10–23, 33–43, 50–64 and 80–87. The structural alteration probably changes the biological function of *w*MUS81. The strong binding ligand to *w*MUS81 is thus able to inhibit the activity of the enzyme.

# 4. Conclusion

In this context, the inhibitory effect of marine compounds to *w*MUS81 was investigated using rigorous computational approaches. Initially, the binding affinity of marine compounds to *w*MUS81 was preliminarily investigated using AutoDock Vina. The binding poses between *w*MUS81 and the 16 top-leads inhibitors were then used as initial conformations of atomistic MD simulations. The binding free energy was calculated via the FEP scheme revealing that five compounds, **D197**, **D220**, **D67**, **D153** and **D547**, are highly potent inhibitors preventing *w*MUS81. Moreover, three compounds, **D152**, **D20** and **D559**, are also good candidate ones, because the predicted inhibition constant of them is in submicromolar affinity ($\Delta G_{FEP}$ is smaller than $-9.50$ kcal mol$^{-1}$). The electrostatic interaction is the dominant factor in the binding of ligands to *w*MUS81.

Over atomistic simulations, it was found that the residues Trp55, Arg59, Leu62, His63 and Arg69 of *w*MUS81 probably play an important role in the binding process of ligands to *w*MUS81 since frequently forming non-bonded contacts and HBs to inhibitors. Especially, Arg59 and Arg69 are the most critical elements governing the binding affinity of ligands to *w*MUS81.

The influence of the ligand **D197**, which formed the largest binding affinity to *w*MUS81, on the conformational ensemble of the protein was also assessed via REMD simulations based on the

hypothesis that the stronger binder will adopt more effect on the stable shape of the protein. Absolutely, the ligand **D197** significantly altered the FEL of wMUS81 and also changed the stable structure of the protein. It may be argued that the ligand **D197**, which forms a strong binding affinity, can change the biological function of wMUS81 due to the change in the enzymic structure.

Data accessibility. The data are available at https://doi.org/10.5061/dryad.m63xsj41r [70], in particular, the docking results (used as the initial structures of MD simulations), MD-refined structures (used as the initial structure for FEP calculations) and REMD-refined structures.

Authors' contributions. S.T.N.: providing an idea, performing MD/FEP/REMD simulations, and drafting the first draft. K.B.V.: statistical analysis of the data. M.Q.P.: preparing the structure of ligands and performing docking simulation. N.M.T.: preparing protein-ligand interaction diagram and SC/HB contacts between them. P.-T.T.: providing an idea and drafting the manuscript.

Competing interests. We declare we have no competing interests.

Funding. This research is funded by Foundation for Science and Technology Development of Ton Duc Thang University (FOSTECT), website: http://fostect.tdtu.edu.vn, under grant no. FOSTECT.2019B.08.

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
