## [Peer Review File · Royal Society Open Science]

Review History

RSOS-210383.R0 (Original submission)

Review form: Reviewer 1

Is the manuscript scientifically sound in its present form?

No

Are the interpretations and conclusions justified by the results?

No

Is the language acceptable?

Yes

Do you have any ethical concerns with this paper?

No

Have you any concerns about statistical analyses in this paper?

No

Recommendation?

Major revision is needed (please make suggestions in comments)

Comments to the Author(s)

In this manuscript, the authors have investigated the potential application of various marine fungi compounds to inhibit the activity of the methyl methanesulfonate and ultraviolet sensitive 81 (wMUS81). The authors establish the high efficacy of eight compounds through molecular docking, MD, REMD and perturbation simulations. Given the importance of MUS81 as a potential cancer drug target and the great biocompatibility of marine products, the study suggests green candidates for altering the enzyme structure and thus activity. However, the manuscript lacks following observations:

1. In section 2.3, the temperature at which MD simulations have been carried out should be mentioned.
3. The REMD study to see the impact of ligands on the structure of receptor has been done only for one compound. On what basis, the particular compound has been chosen. Also, the structural changes induced by only one compound cannot be generalized for all. The study should be done with some other ligands as well to establish a generalized view (structure-property relationship).
2. The MD section has also been discussed focusing the binding efficacy of the compounds to the receptor. The trajectories can further be analyzed to see how the structure around the binding site is changed to cause the inactivation of the enzyme.
3. The axes of contour plots in Figure 7 are not labelled.
4. The writing style is quite confusing and must be improved. Also, the conclusion is a simple repetition of lines from discussion part.

Review form: Reviewer 2

Is the manuscript scientifically sound in its present form?

No

Are the interpretations and conclusions justified by the results?

No

Is the language acceptable?

No

Do you have any ethical concerns with this paper?

No

Have you any concerns about statistical analyses in this paper?

Yes

Recommendation?

Reject

Comments to the Author(s)

The manuscript by Ngo et al. is primarily a report on the docking of a library of compounds on the protein wMUS81, a DNA endonuclease, and an evaluation of their binding strengths. The purported aim of the study is "to be able to prevent wMUS81 structure". However, I found the manuscript lacking in description of) motivation, technical rigor, and discussion.

The aim of the manuscript is unclear. Do the authors intend to distort the natively folded form of wMUS81 and thereby alter its downstream biological function?

On what basis were the eight compounds chosen? Was the basis clinical or structural? I could not find a tabulated summary of the molecules.

I do not understand the necessity of evaluating the binding free energies via docking and from the MD snapshots; the latter ought to be more rigorous. However, the MD simulation sampling is suboptimal.

The legends of Figures and Tables are not descriptive enough. Also, energy units are not added where necessary.

I found almost no mechanistic insights into the ligand-protein binding, and am unable to appreciate how this study may be useful to the community. Thereby, I am unable to recommend the manuscript for publication.

Decision letter (RSOS-210383.R0)

Dear Dr Ngo:

Manuscript ID: RSOS-210383

Title: "Marine derivatives prevent wMUS81 in silico studies"

Thank you for submitting the above manuscript to Royal Society Open Science. Your paper was sent to reviewers and their comments are included at the bottom of this letter.

In view of the concerns raised by the reviewers, the manuscript has been rejected in its current form. However, a new manuscript may be submitted which takes into consideration these comments.

Please note that resubmitting your manuscript does not guarantee eventual acceptance, and that your resubmission will be subject to peer review before a decision is made.

Your resubmitted manuscript should be submitted by 09-Nov-2021. If you are unable to submit by this date please contact the Editorial Office.

Royal Society of Chemistry
Thomas Graham House
Science Park, Milton Road

Cambridge, CB4 0WF
Royal Society Open Science - Chemistry Editorial Office

On behalf of the Subject Editor Professor Anthony Stace and the Associate Editor Dr Debashree Ghosh

REVIEWER(S) REPORTS:

Associate Editor Comments to Author ():

RSC Associate Editor:

Comments to the Author:

The reviews raise several issues. However, the most major scientific issue that they mention is that the importance and motivation of the paper is unclear and the conclusion is also not well written. Due to these reasons the paper cannot be accepted in its current form. However, if the authors can address the issues raised with a point by point reply, it may be re-evaluated.

RSC Subject Editor:

Comments to the Author:

(There are no comments.)

Reviewers' Comments to Author:

Reviewer: 1

Comments to the Author(s)

In this manuscript, the authors have investigated the potential application of various marine fungi compounds to inhibit the activity of the methyl methanesulfonate and ultraviolet sensitive 81 (wMUS81). The authors establish the high efficacy of eight compounds through molecular docking, MD, REMD and perturbation simulations. Given the importance of MUS81 as a potential cancer drug target and the great biocompatibility of marine products, the study suggests green candidates for altering the enzyme structure and thus activity. However, the manuscript lacks following observations:

1. In section 2.3, the temperature at which MD simulations have been carried out should be mentioned.
3. The REMD study to see the impact of ligands on the structure of receptor has been done only for one compound. On what basis, the particular compound has been chosen. Also, the structural changes induced by only one compound cannot be generalized for all. The study should be done with some other ligands as well to establish a generalized view (structure-property relationship).
2. The MD section has also been discussed focusing the binding efficacy of the compounds to the receptor. The trajectories can further be analyzed to see how the structure around the binding site is changed to cause the inactivation of the enzyme.
3. The axes of contour plots in Figure 7 are not labelled.
4. The writing style is quite confusing and must be improved. Also, the conclusion is a simple repetition of lines from discussion part.

Reviewer: 2

Comments to the Author(s)

The manuscript by Ngo et al. is primarily a report on the docking of a library of compounds on the protein wMUS81, a DNA endonuclease, and an evaluation of their binding strengths. The

purported aim of the study is “to be able to prevent wMUS81 structure”. However, I found the manuscript lacking in description of) motivation, technical rigor, and discussion.

The aim of the manuscript is unclear. Do the authors intend to distort the natively folded form of wMUS81 and thereby alter its downstream biological function?

On what basis were the eight compounds chosen? Was the basis clinical or structural? I could not find a tabulated summary of the molecules.

I do not understand the necessity of evaluating the binding free energies via docking and from the MD snapshots; the latter ought to be more rigorous. However, the MD simulation sampling is suboptimal.

The legends of Figures and Tables are not descriptive enough. Also, energy units are not added where necessary.

I found almost no mechanistic insights into the ligand-protein binding, and am unable to appreciate how this study may be useful to the community. Thereby, I am unable to recommend the manuscript for publication.

Author's Response to Decision Letter for (RSOS-210383.R0)

See Appendix A.

RSOS-210974.R0

Review form: Reviewer 1

Is the manuscript scientifically sound in its present form?

Yes

Are the interpretations and conclusions justified by the results?

Yes

Is the language acceptable?

Yes

Do you have any ethical concerns with this paper?

No

Have you any concerns about statistical analyses in this paper?

No

Recommendation?

Accept as is

Comments to the Author(s)

The authors have addressed most of the concerns I had on the original manuscript. The revised version can now be accepted for publication.

Review form: Reviewer 2

Is the manuscript scientifically sound in its present form?

No

Are the interpretations and conclusions justified by the results?

Yes

Is the language acceptable?

No

Do you have any ethical concerns with this paper?

No

Have you any concerns about statistical analyses in this paper?

No

Recommendation?

Major revision is needed (please make suggestions in comments)

Comments to the Author(s)

The manuscript presents efforts to discover potential small molecule candidates for inhibition of the activity of the winged-helix domain of the methyl methanesulfonate ultraviolet sensitive 81 (wMUS81), a protein associated with disease. The authors begin by screening and docking hundreds of naturally synthesized small organic molecules, followed by identification of those that yield the most significant binding scores. The docked poses of the latter category were then simulated atomistically, and the commonly interacting residues of wMUS81 identified. Principal component analysis of one putative docked entity was performed and compared with the unbound protein. Presumptively, altered conformational propensities will prevent the pathogenic effects of wMUS81. While the results are not accompanied with experimental validation, the results could be of value in drug design. However, the authors should address the following concerns prior to reconsideration for publication:

1. Describe the docking process in detail. Was the docking process for the 563 ligands automated? Was the docking rigid or flexible? Were any user-specific parameters or bounds used? What were the "limitations" mentioned on Pg. 8?

2. MD simulations and Free Energy calculations:

- a. Why were 4 Na⁺ ions "approximately" used to neutralize the system?
- b. It's not clear how the authors determine the volume of the ligand as ca. 74 nm³.
- c. REMD simulations: what was the total data generated (in nanoseconds), the acceptance ratio, and the net data used for the analysis?
- d. Details of the FEP calculations should be briefly included in Methods.
- e. Why were the solvation free energy components ignored in the net binding free energy?
- f. In addition to the free energy landscape constructed from the 1st and 2nd principal components, more intuitive collective variables should be used, particularly radius of gyration and number of internal native contacts of the protein. The differences arising due to the ligand binding should be discussed.

3. The assertion that "the residues Arg59, His63, and Arg69 not only form hydrogen bond contacts with the inhibitors but also adopts π -cation/ π - π contacts to the ligands" is not based on rigorous analysis. The authors should either qualify or modify this statement.

4. The manuscript uses non-standard English language in several places that makes comprehension difficult for the reader. A careful, third-party proofreading is highly recommended.

Decision letter (RSOS-210974.R0)

Dear Dr Ngo:

Title: Marine derivatives prevent wMUS81 in silico studies

Manuscript ID: RSOS-210974

Thank you for submitting the above manuscript to Royal Society Open Science. On behalf of the Editors and the Royal Society of Chemistry, I am pleased to inform you that your manuscript will be accepted for publication in Royal Society Open Science subject to minor revision in accordance with the referee suggestions. Please find the reviewers' comments at the end of this email.

The reviewers and handling editors have recommended publication, but also suggest some minor revisions to your manuscript. Therefore, I invite you to respond to the comments and revise your manuscript.

Because the schedule for publication is very tight, it is a condition of publication that you submit the revised version of your manuscript before 05-Aug-2021. Please note that the revision deadline will expire at 00.00am on this date. If you do not think you will be able to meet this date please let me know immediately.

- 1) A text file of the manuscript (tex, txt, rtf, docx or doc), references, tables (including captions) and figure captions. Do not upload a PDF as your "Main Document".
- 2) A separate electronic file of each figure (EPS or print-quality PDF preferred (either format should be produced directly from original creation package), or original software format)
- 3) Included a 100 word media summary of your paper when requested at submission. Please ensure you have entered correct contact details (email, institution and telephone) in your user account

- 4) Included the raw data to support the claims made in your paper. You can either include your data as electronic supplementary material or upload to a repository and include the relevant doi within your manuscript
- 5) All supplementary materials accompanying an accepted article will be treated as in their final form. Note that the Royal Society will neither edit nor typeset supplementary material and it will be hosted as provided. Please ensure that the supplementary material includes the paper details where possible (authors, article title, journal name).

Kind regards,
Dr Laura Smith
Publishing Editor, Journals

On behalf of the Subject Editor Professor Anthony Stace and the Associate Editor Dr Debashree Ghosh.

RSC Associate Editor

Comments to the Author:

The manuscript can be accepted after the authors address all the issues pointed out by the referees and provide a point by point reply of the changes in the manuscript.

Reviewer comments to Author:

Reviewer: 2

Comments to the Author(s)

The manuscript presents efforts to discover potential small molecule candidates for inhibition of the activity of the winged-helix domain of the methyl methanesulfonate ultraviolet sensitive 81 (wMUS81), a protein associated with disease. The authors begin by screening and docking hundreds of naturally synthesized small organic molecules, followed by identification of those that yield the most significant binding scores. The docked poses of the latter category were then simulated atomistically, and the commonly interacting residues of wMUS81 identified. Principal component analysis of one putative docked entity was performed and compared with the

unbound protein. Presumptively, altered conformational propensities will prevent the pathogenic effects of wMUS81. While the results are not accompanied with experimental validation, the results could be of value in drug design. However, the authors should address the following concerns prior to reconsideration for publication:

1. Describe the docking process in detail. Was the docking process for the 563 ligands automated? Was the docking rigid or flexible? Were any user-specific parameters or bounds used? What were the "limitations" mentioned on Pg. 8?
2. MD simulations and Free Energy calculations:
 - a. Why were 4 Na⁺ ions "approximately" used to neutralize the system?
 - b. It's not clear how the authors determine the volume of the ligand as ca. 74 nm³.
 - c. REMD simulations: what was the total data generated (in nanoseconds), the acceptance ratio, and the net data used for the analysis?
 - d. Details of the FEP calculations should be briefly included in Methods.
 - e. Why were the solvation free energy components ignored in the net binding free energy?
 - f. In addition to the free energy landscape constructed from the 1st and 2nd principal components, more intuitive collective variables should be used, particularly radius of gyration and number of internal native contacts of the protein. The differences arising due to the ligand binding should be discussed.
3. The assertion that "the residues Arg59, His63, and Arg69 not only form hydrogen bond contacts with the inhibitors but also adopts π -cation/ π - π contacts to the ligands" is not based on rigorous analysis. The authors should either qualify or modify this statement.
4. The manuscript uses non-standard english language in several places that makes comprehension difficult for the reader. A careful, third-party proofreading is highly recommended.

Reviewer: 1

Comments to the Author(s)

The authors have addressed most of the concerns I had on the original manuscript. The revised version can now be accepted for publication.

Author's Response to Decision Letter for (RSOS-210974.R0)

See Appendix B.

Decision letter (RSOS-210974.R1)

Dear Dr Ngo:

Title: Marine derivatives prevent wMUS81 in silico studies

Manuscript ID: RSOS-210974.R1

It is a pleasure to accept your manuscript in its current form for publication in Royal Society Open Science. The chemistry content of Royal Society Open Science is published in collaboration with the Royal Society of Chemistry.

Yours sincerely,
Dr Ellis Wilde
Publishing Editor, Journals

On behalf of the Subject Editor Professor Anthony Stace and the Associate Editor Dr Debashree Ghosh.

RSC Associate Editor
Comments to the Author:
(There are no comments.)

Reviewer(s)' Comments to Author:

Appendix A

TON DUC THANG UNIVERSITY
19 Nguyen Huu Tho, Ho Chi Minh City, Vietnam

Son Tung Ngo, PhD

Head of Theoretical and Computational Biophysics Laboratory

ngosontung@tdtu.edu.vn

0084 93 1771 747

Ho Chi Minh City, Jun 3, 2020

Editor of RSC Advances

Manuscript: Marine derivatives prevent wMUS81 *in silico* studies

Manuscript ID: RSOS-210383 R1

Author(s): Son Tung Ngo, Khanh B. Vu, Minh Quan Pham, Nguyen Minh Tam, Phuong-Thao Trang

Dear Dr Laura Smith

Editor of Royal Society Open Science,

We sincerely thank two reviewers for their comments to enhance the impact of the manuscript. We have carefully considered the review reports and accordingly revised the manuscript. Our point-to-point responses and corrections are summarized as follows:

Reviewers' Comments to Author:

Reviewer: 1

Comments:

In this manuscript, the authors have investigated the potential application of various marine fungi compounds to inhibit the activity of the methyl methanesulfonate and ultraviolet sensitive 81 (wMUS81). The authors establish the high efficacy of eight compounds through molecular docking, MD, REMD and perturbation simulations. Given the importance of MUS81 as a potential cancer drug target and the great biocompatibility of marine products, the study suggests green candidates for altering the enzyme structure and thus activity. However, the manuscript lacks following observations:

Our reply: We would like to thank the reviewer for the recommendation.

Comment 1: In section 2.3, the temperature at which MD simulations have been carried out should be mentioned.

Our reply: Thanks for your suggestion. The MD temperature is of 310 K and the information was inserted into the main texts with red color highlights.

Comment 2: The REMD study to see the impact of ligands on the structure of receptor has been done only for one compound. On what basis, the particular compound has been chosen. Also, the structural changes induced by only one compound cannot be generalized for all. The study should be done with some other ligands as well to establish a generalized view (structure-property relationship).

Our reply: Thanks for your comments. It should be noted that the stronger ligand-binding to protein meaning that the better inhibition ref. 26, 64. Moreover, the stronger binder normally adopts a strong effect on the protein, resulting in the structural change of protein was recorded ref. 52, 64. Therefore, in this context, the influence of the ligand D197 on the structure of wMUS81 was investigated using REMD simulations since the ligand formed the largest binding affinity to wMUS81 via FEP calculations. In addition, due to a huge computing resource was required for REMD simulation, we have only chosen the largest binding affinity compound for structural changing study. Additional discussion was inserted into the main texts with red color highlight (page. 10).

Comment 3: The MD section has also been discussed focusing the binding efficacy of the compounds to the receptor. The trajectories can further be analyzed to see how the structure around the binding site is changed to cause the inactivation of the enzyme.

Our reply: Thanks for your suggestion. As mentioned above, stronger binder means the better inhibition, therefore, we have investigated the structural change of wMUS81 under effect of the ligand **D197**, which formed largest binding affinity, by using REMD simulations.

Comment 4: The axes of contour plots in Figure 7 are not labelled.

Our reply: The axes title was inserted as your recommendation.

Comment 5: The writing style is quite confusing and must be improved. Also, the conclusion is a simple repetition of lines from discussion part.

Our reply: Thanks for your comments. The manuscript was carefully checked and improved.

Reviewer: 2

Comments:

The manuscript by Ngo et al. is primarily a report on the docking of a library of compounds on the protein wMUS81, a DNA endonuclease, and an evaluation of their binding strengths. The purported aim of the study is “to be able to prevent wMUS81 structure”. However, I found the manuscript lacking in description of) motivation, technical rigor, and discussion.

Our reply: We thank the reviewer for the comments and the answers are as follow.

Comments 1: The aim of the manuscript is unclear. Do the authors intend to distort the natively folded form of wMUS81 and thereby alter its downstream biological function?

Our reply: Thanks for your comments. We have revised the introduction part (red color highlight in page 3) in order to clarify the issue, in which the work aims to screening possible compounds for inhibiting wMUS81. The stronger binding ligand probably is the better inhibition. Besides, the REMD simulations were carried out to clarify the influence of the strongest binder on structural change of wMUS81.

Comments 2: On what basis were the eight compounds chosen? Was the basis clinical or structural? I could not find a tabulated summary of the molecules.

Our reply: Thanks for your comment but the reviewer probably misunderstood this part. As mentioned in the manuscript, the ligands were marine fungi compounds, which obtained in the previous work (ref. 39-

41) and *Cao et al* unpublished work. The list of compounds were mentioned in the Supporting information in details.

Comment 3: I do not understand the necessity of evaluating the binding free energies via docking and from the MD snapshots; the latter ought to be more rigorous. However, the MD simulation sampling is suboptimal.

Our reply: Thanks for your comment. The reviewer probably misunderstood this part. As mentioned in the Introduction Section, the molecular docking simulations were used to preliminarily predict the ligand-binding affinity and pose. However, because molecular docking approach normally uses much constraints to enhance the computing speed, the docking results were limited. The MD simulations are thus employed to refine the docking observation. In particular, the MD-refined binding pose often close to the native binding pose rather than that from docking simulation. The MD results are thus more consistent with the experiment than the docking one. The comparison of docking and MD results imply the suitability of the docking approach. If the docking result is too far from the MD-refined result, a poor correlation between MD and docking simulations was thus observed. However, a good correlation coefficient was obtained implying that Vina is an appropriate protocol to characterize the binding affinity and binding pose of ligands to *w*MUS81. Applying Vina to screen a large database of compounds to *w*MUS81 is appropriate. Moreover, as mentioned in the manuscript and Supporting information, the complex structures reached the equilibrium states after the half of MD trajectories and two trajectories of MD simulations were generated for each systems. The MD simulation sampling is thus appropriate.

Comment 4: The legends of Figures and Tables are not descriptive enough. Also, energy units are not added where necessary.

Our reply: Thanks for your comment. The legends of Figures and Tables were checked and fixed.

Comment 5: I found almost no mechanistic insights into the ligand-protein binding, and am unable to appreciate how this study may be useful to the community. Thereby, I am unable to recommend the manuscript for publication.

Our reply: In this work, the binding free energy of ligands to *w*MUS81 was carefully investigated using rigorous computational methods. In particular, the critical residues influencing the binding process were revealed. As well as, the binding free energy terms were investigated suggesting that the electrostatics plays an important role in the binding of ligands to *w*MUS81. The strongest binder can also inhibit the biological activity of *w*MUS81 via alter the structure of the enzyme. The observation should contribute to the exist knowledge.

Overall, we have addressed all concerns of all reviewers and submitted a revised manuscript.

We thank the reviewers and the editor for helpful comments and suggestions.

Son Tung Ngo

Appendix B

TON DUC THANG UNIVERSITY
19 Nguyen Huu Tho, Ho Chi Minh City, Vietnam

Son Tung Ngo, PhD

Head of Theoretical and Computational Biophysics Laboratory

ngosontung@tdtu.edu.vn

0084 93 1771 747

Ho Chi Minh City, Jun 3, 2020

Editor of RSC Advances

Manuscript: Marine derivatives prevent wMUS81 *in silico* studies

Manuscript ID: RSOS-210383 R1

Author(s): Son Tung Ngo, Khanh B. Vu, Minh Quan Pham, Nguyen Minh Tam, Phuong-Thao Trang

Dear Dr Laura Smith

Editor of Royal Society Open Science,

We sincerely thank the reviewer 1 for his/her recommend and the reviewer 2 for his/her comments to enhance the impact of the manuscript. We have carefully considered the review reports and accordingly revised the manuscript. Our point-to-point responses and corrections are summarized as follows:

Reviewers' Comments to Author:

Reviewer: 1

Comments:

The authors have addressed most of the concerns I had on the original manuscript. The revised version can now be accepted for publication.

Our reply: We would like to thank the reviewer for the recommendation.

Reviewer: 2

Comments:

The manuscript presents efforts to discover potential small molecule candidates for inhibition of the activity of the winged-helix domain of the methyl methanesulfonate ultraviolet sensitive 81 (wMUS81), a protein associated with disease. The authors begin by screening and docking hundreds of naturally synthesized small organic molecules, followed by identification of those that yield the most significant binding scores. The docked poses of the latter category were then simulated atomistically, and the commonly interacting residues of wMUS81 identified. Principal component analysis of one putative docked entity was performed and compared with the unbound protein. Presumptively, altered conformational propensities will prevent the pathogenic effects of wMUS81. While the results are not accompanied with experimental validation, the results could be of value in drug design. However, the authors should address the following concerns prior to reconsideration for publication:

Our reply: We thank the reviewer for the comments and the answers are as follow.

Comment 1: Describe the docking process in detail. Was the docking process for the 563 ligands automated? Was the docking rigid or flexible? Were any user-specific parameters or bounds used? What were the “limitations” mentioned on Pg. 8?

Our reply: All of docking parameters was reported in the manuscript. Moreover, during the docking simulations, the ligand was flexible but the receptor was fully rigid. Furthermore, the limitations of the docking approach includes a limited number of trial ligand positions, employed united-atoms force field, and utilizing implicit solvent model, etc. Additional discussion was inserted into the manuscript with red color highlight.

Comment 2: MD simulations and Free Energy calculations: Why were 4 Na⁺ ions “approximately” used to neutralize the system?

Our reply: There are some ligands having different charges since different in protonation states.

Comment 3: MD simulations and Free Energy calculations: It’s not clear how the authors determine the volume of the ligand as ca. 74 nm³.

Our reply: The volume of system was reported by GROMACS when the “gmx editconf” was used to generate the PBC box.

Comment 4: REMD simulations: what was the total data generated (in nanoseconds), the acceptance ratio, and the net data used for the analysis?

Our reply: Thanks for your comment. The REMD simulations were carried out with with a length of 100 ns each replica (6400 ns for two systems totally). The acceptance ratio was 15 % approximately. The first 40 ns of REMD simulations were removed from any analysis to evade the initial biased. Additional information was inserted into the manuscript with red color highlight.

Comment 5: Details of the FEP calculations should be briefly included in Methods.

Our reply: Thanks for your comment. Additional information was inserted into the main text with red color highlight.

Comment 6: Why were the solvation free energy components ignored in the net binding free energy?

Our reply: Thanks for your comment. As mentioned in the materials and methods, the binding free energy was calculated by the free energy perturbation method. In particular, the coupling parameter λ was used to modify the Hamiltonian of the system from bound state to unbound state. The free energy change of the process was summed using the Bennet acceptance ratio method (ref. 60). λ -modification simulations were thus performed to calculate the free energy change of ligand-annihilation from the solvated complex and ligand systems. The difference of free energy change of two processes is the binding free energy of ligands to receptors. During which, the free energy change over the ligand-annihilation from the solvate ligand system is the de-solvation free energy of ligands.

Comment 7: In addition to the free energy landscape constructed from the 1st and 2nd principal components, more intuitive collective variables should be used, particularly radius of gyration and number of internal native contacts of the protein. The differences arising due to the ligand binding should be discussed.

Our reply: Thanks for your comment. The free energy landscape was constructed by PCA methods. It is a standard approach considering the structural change of biomolecules during atomistic simulations, although the collective-variable FEL probably provides insight into the folding of a protein. Here, we considered the structural change of the complexes, therefore the PCA method should be more suitable than the collective-variable FEL.

Comment 8: The assertion that “the residues Arg59, His63, and Arg69 not only form hydrogen bond contacts with the inhibitors but also adopts π -cation/ π - π contacts to the ligands” is not based on rigorous analysis. The authors should either qualify or modify this statement.

Our reply: Thanks for your comment. The discussion was based on the docking pose analysis over 16 complexes. The further analysis on MD-refined structure were then performed and discussed below. The additional description for figure 5 was inserted to clarify the issue.

Comment 9: The manuscript uses non-standard english language in several places that makes comprehension difficult for the reader. A careful, third-party proofreading is highly recommended.

Our reply: Thanks for your suggestion. The manuscript was carefully checked and edited.

Overall, we have addressed all concerns of all reviewers and submitted a revised manuscript.

We thank the reviewers and the editor for helpful comments and suggestions.

Son Tung Ngo